# Optimization of Applied Irrigation Water for High Marketable Yield, Fruit Quality and Economic Benefits of Processing Tomato Using a Low-Cost Wireless Sensor

Antonio El Chami [1], Raffaele Cortignani [1], Davide Dell'Unto [1], Roberto Mariotti [2], Piero Santelli [3], Roberto Ruggeri [1], Giuseppe Colla [1] and Mariateresa Cardarelli [1,*]

1. Dipartimento di Scienze Agrarie e Forestali, Università della Tuscia, 01100 Viterbo, Italy
2. Agenzia Regionale per lo Sviluppo e l'Innovazione dell'Agricoltura del Lazio, 01016 Tarquinia, Italy
3. TORO System Europe Srl, 00065 Fiano Romano, Italy
* Correspondence: tcardare@unitus.it

**Abstract:** Water management is a key factor to optimize fruit quality and yield of processing tomatoes which are site-specific and influenced by environmental conditions e.g., soil, temperature, precipitation. The aim of this study was to evaluate the efficacy of a low-cost wireless soil moisture sensor in determining the irrigation level for optimizing the marketable yield, fruit quality and economic profit of processing tomato. A two-years (2017–2018) trial was conducted in open field, applying nine drip irrigation levels controlled by wireless soil moisture capacitance sensors. The irrigation levels were as follows: 13.2, 16.7, 25.4, 33.3, 50.0, 62.3, 82.5, 100 and 186.8% of water restitution based on soil moisture sensor readings. Because of the crop stress induced by heavy rainfalls occurring in 2018 growing season, total and marketable yields reached higher maximum values in 2017 than 2018. In 2017, total and marketable yields were maximized by supplying 92.8% and 96.2% of irrigation level, respectively. Moreover, 95.6% and 91.2% of irrigation level were necessary in 2018 to maximize total and marketable yield, respectively. In both growing seasons, marketable yield variation was due to changes of both fruit number and fruit mean weight. Total soluble solids of fruit juice linearly decreased by increasing the irrigation level with a more pronounced effect in the driest growing season (2017). Economic analysis demonstrated that 100% of irrigation level should be preferred by the Italian farmers since it maximized the operating margins of processing tomatoes in both years. To conclude, the use of the tested low-cost wireless soil moisture sensor is an effective tool to manage the level of irrigation and optimize the processing tomato yield and economic benefits for farmers.

**Keywords:** water management; processing tomato; soil moisture sensor; marketable yield; fruit quality; total soluble solids; operational margin

## 1. Introduction

Water scarcity is one of the priority challenges of last century. Agriculture plays a fundamental role as it is responsible for 70% of global water withdrawals, continuously stressing freshwater resources. While global water use and food production demand are growing, climate changes affect agricultural productions with severe drought events and constant temperature increases, resulting in production loss and greater crops water requirements [1]. Moreover, irrigated agriculture often lacks in efficiency, leading to considerable amount of water wasting during storage, transport, and field application [2,3]. "Water-saving agriculture" is a complex system involving agronomic and hydraulic engineering techniques in the integrated exploitation of water, soil, and crop resources. This concept implies an intervention on the different steps of the irrigation process: (1) management and maintenance of sources; (2) use of updated irrigation techniques and field management; (3) use of agronomic measures and biological techniques to maximize specific crop water use efficiency [4].

To optimize irrigation, several methods are available, and a variety of irrigation control devices are on the market [5]. Despite its widely use, the water balance (crop water requirement or evapotranspiration modelling) approach could be not very accurate [6]. Meteorological parameters used for potential evapotranspiration ($ET_0$) calculation, in commercial vegetable production, are often not representative of the microclimatic conditions of the growing field due to small-scale variations caused by orographic conditions. Considerable variability in field pedo-climatic conditions could occur in terms of times and space and the lack of real-time information for a fast and continuous integration needs of the modelling approach is certainly the method weakness. Moreover, imprecise crop coefficients and time-consuming elaboration makes the modelling approach, for irrigation scheduling purpose, not always easy to be applied at farm level [7].

Soil moisture monitoring is another way to manage the irrigation of crops. However, the use of this approach has been limited by cost and labor for sensor installation. Nowadays a diversity of dielectric sensors, using time domain reflectometry [8] or frequency domain reflectometry technologies [9], are available, unexpensive and integrated in practical devices making them a user-friendly tool for irrigation management. This class of sensors measures the soil bulk permittivity (or dielectric constant), which controls the speed of an electromagnetic wave or pulse through the soil [10]. The use of sensors to monitor the soil water status, enables irrigation scheduling to be more precise and dynamic with high level of automation [11] providing higher accuracy in comparison with modelling approaches [12]. Soil moisture content is maintained within a predefined upper and lower threshold, through the period of the growing season, thanks to sensor-based irrigation scheduling [13].

Processing tomato (*Solanum lycopersicum* L.) is a widespread crop with high water demand [14]. Most of the world processing tomato acreage is located in arid and semiarid regions [15], often exposed to drought and high temperature, thus requiring irrigation throughout the growing season. Mediterranean productions involve high irrigation volume inputs, particularly in southern regions where crops water requirement could reach peaks of 800 mm in hottest growing areas and seasons [16]. Therefore, a strong effect of irrigation regime on yield and quality of processing tomatoes has been extensively reported especially in dry areas [14,17,18]. Usually, limited soil water availability resulted in a decrease of tomato yield with significant changes in fruit quality [19]. Increasing in vitamin C content, dry matter and soluble solids of the fruits were recorded during water shortage conditions [20]. In some countries such as Italy, the decrease of marketable yield associated with drought conditions is not necessarily associated with a reduction of farmer income because sale price is also determined by the total soluble solids in the fruits with a premium price for high-quality tomatoes. Whitin this context, many deficit irrigation approaches were developed to optimize irrigation to achieve better results in terms of water saving, production and quality of processing tomato in dry areas [15,19,21–25].

Some authors obtained better results in terms of water saving quality and biomass production in processing tomatoes with soil moisture measurement method compared with ACQUACROP (FAO) modelling approach, which is based on crop evapotranspiration [26]. A capacitance soil moisture sensor was used for scheduling irrigation in processing tomato grown in Spain without significant differences in terms of marketable yield compared to ET-based method and granular matrix sensor [27]. However, this study was limited to one irrigation level and one-growing cycle.

The objective of this work was to study the impact of irrigation level on processing tomatoes in two growing cycles (2017 and 2018) using a low-cost wireless soil moisture capacitance sensor (Precision™ Soil Sensor–Toro, Bloomington, MN, USA) able to accurately measure soil moisture content up to $4\,\mathrm{dS\,m^{-1}}$ of salinity; this sensor was successfully tested in tall fescue and bermudagrass [28]. In the current study the irrigation effects were evaluated from agronomic and economic point of view.

## 2. Materials and Methods

### 2.1. Study Site

The trial was conducted at the Experimental Farm of Agenzia Regionale per lo Sviluppo e l'Innovazione dell'Agricoltura del Lazio located in Portaccia (Tarquinia) (coordinates c. $42°13'47''$ N, $11°43'23''$ E; 22 m a.s.l.), in the rural district of Tuscia (Viterbo, Latium Region, Italy). The area is subjected to a Mediterranean inferior mesothermic dry climate and is in a thermo-Mediterranean and meso-Mediterranean subregion belonging to the xerothermic region. The Experimental Farm includes an agro-meteorological station measuring meteorological data such as temperature and rain; moreover, the agro-metereological station included a Stainless Steel Class A Evaporation Pan sited on a green grass cover and the estimation of $ET_0$ was obtained using the Pan evaporation method proposed by FAO.

The soil was planted by durum wheat before both growing cycles of processing tomato. The soil contained 48% sand, 18% silt, and 34% clay. Soil chemical characteristics were: pH 6.8, electrical conductivity of saturated paste extract 0.151 dS/m, organic matter 1.17%, total nitrogen 0.077%, cation exchange capacity 19.9 meq/100 g, organic carbon 0.68%, exchangeable cations were as follow (mg/kg): K (617), P (57), Ca (2820), Mg (299), Na (83). Soil trace elements were as follow (mg/kg): soluble B (0.98), assimilable Fe (28.6), assimilable Mn (28.2), assimilable Cu (2.0), assimilable Zn (0.8). Based on the physico-chemical analysis, the soil was classified as sandy clay loamy, with neutral pH, medium cation exchange capacity, normal salinity, and low organic matter content; moreover, the soil was very high in exchangeable K, Ca, and Mg, and available P, high in available Mn and Fe, medium in soluble B, available Cu, and low in exchangeable Na, available Zn, and total N.

### 2.2. Cropping Details

Processing tomato (*Solanum lycopersicum* L.-Perfectpeel ex PS 1296, Bayer Seminis Milano, Italy) was grown in two growing cycles (2017 and 2018). Transplants at 2–3 true leaves stage were planted in double rows (distance between double rows: 1.7 m; distance between rows: 0.5 m; plant distance along the row: 0.36 m) at a plant density of 3.2 plants/m$^2$ on 15 May 2017 and 28 May 2018. A slow-release mineral fertilizer containing 16N-9.6P-13.2K was pre-plant applied at a rate of 1 t/ha followed by two side dress applications during flowering of a granular ammonium nitrate fertilizer (26%N) at a rate of 76.9 kg/ha for each application. Granular fertilizer was mixed into the upper portion of the soil just after side dress application. Pests and diseases were controlled as needed using copper-based fungicides, and abamectin and deltamethrin-based insecticides at the label rate. Weeds were controlled by hand hoeing.

### 2.3. Experimental Design and Treatments

In both years, nine irrigation treatments were tested in a randomized block design with 3 replicates. The irrigation treatments were obtained using 9 drip lines having different water flow rates as follow: 1.5, 1.9, 2.9, 3.8, 5.7, 7.1, 9.4, 11.4 and 21.3 L/m h. In all plots, irrigation was scheduled following the readings of soil moisture sensors (Precision™ Soil Sensor–Toro, Bloomington, MN, USA) located in plot with the drip lines having a flow rate of 11.4 L/m h (Soil moisture sensor treatment or SMS treatment). Just after transplanting, field capacity was determined using a built-in calibration process as proposed by Toro Company and used in other research studies [28]. Following the manufacturer's instructions, near-saturation soil conditions were created by running two cycles of irrigation and wetting the area near the sensors and then, the calibration process was initiated. The soil moisture sensor system defined field capacity based on the soil moisture values recorded after 4 h of drainage following the saturation event [28]. When the soil water content in the plot of SMS treatment was lower than 90% of the field capacity, the irrigation was activated in all plots until the probe reading reached 100% (field capacity).Therefore, field capacity was maintained during the growing cycles in the three plots of SMS treatment (100% of water restitution based on SMS readings) while soil water depletion was expected in plots

having drip lines with flow rates lower than 11.4 L/m h as for 1.5 (13.2% of SMS treatment), 1.9 (16.7% of SMS treatment), 2.9 (25.4% of SMS treatment), 3.8 (33.3% of SMS treatment), 5.7 (50.0% of SMS treatment), 7.1 (62.3% of SMS treatment), 9.4 (82.5% of SMS treatment). Treatment having drip lines with flow rate of 21.3 L/m h provided 186.8% more water than SMS treatment (11.4 L/m h). A total of 27 plots were set up in each growing cycle. Each plot contained three double rows of 20 plants each.

The SMS (Precision™ Soil Sensor – Toro, Bloomington, MN, USA) was a wireless time domain reflectometry sensor that measured the electrical permittivity of the soil environment providing measurements of soil moisture values [29]. Each SMS was placed in the middle of two plants along the row at a depth of 10 cm (Figure 1); the SMS was connected to a control unit for automation of irrigation scheduling. The cost of the sensor including one control unit was about 250 €/unit (https://pratoerboso.com/it; accessed on 5 March 2023). In all treatments, irrigation was scheduled as a function of the soil water readings of the probe positioned in the plot having drip lines with a flow rate of 11.4 L/m h (SMS treatment).

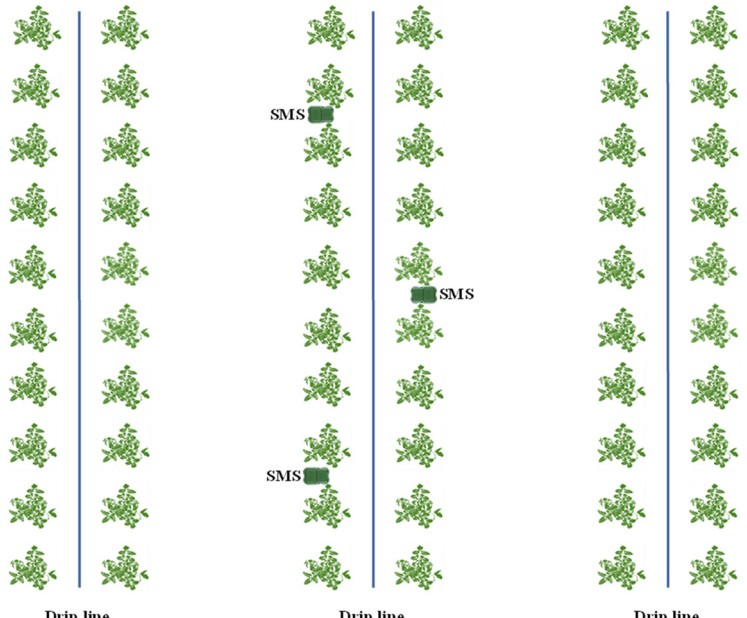

**Figure 1.** Location of wireless soil moisture sensors (SMS) halfway between plants along the central row of the plot.

### 2.4. Growing Degree Days, Production and Fruit Quality

The accumulated values of growing degree days (GDD) from transplanting to harvest were calculated for both growing cycles using the formula proposed by [30] as follow:

$$GDD = \frac{(T_{Max} + T_{Min})}{2} - T_{Base} \tag{1}$$

$T_{Base}$ is the minimum developmental threshold temperature which was fixed for tomato to 10 °C. If the daily minimum temperature ($T_{Min}$) was below the $T_{Base}$, the $T_{Min}$ was set equal to $T_{Base}$ in the formula [30]. This modification assumes that tomato plant growth is reduced when temperatures are below 10 °C and it helps avoid GDD from being negative for minimum temperatures below freezing. If daily maximum temperature ($T_{Max}$) was above 35 °C, it was reset to $T_{Cutoff}$ in the formula [30]. $T_{Cutoff}$ was derived from the following equation:

$$T_{Cutoff} = 35 - 2(T_{Max} - 35) \tag{2}$$

This modification of T$_{Max}$ assumes that tomato plant growth slows down as leaf stomates close and cellular respiration increases when temperature begins to exceed 35 °C.

Agronomic effects of the irrigation treatments on tomatoes were evaluated considering quali-quantitative fruit traits such as marketable yield, unmarketable yield, total yield, percentage of marketable fruits, percentage of rotten fruits, marketable fruit number, marketable fruit mean weight, fruit firmness, percentage of marketable, immature, and rotten fruits, and total soluble solids of fruit juice.

In both growing cycles, fruit harvest starts when the visual estimation of mature fruits on the total reached about 90%. At harvest (14/08/2017-92 DAT; 28/08/2018-93 DAT), fruits were collected separately in the central double row of each plot, and sorted in marketable (red and orange-red fruits), immature (light orange, and green fruits), and rotten fruits. Fruits of three groups were weighted in each plot. Tomato fruits that were decayed or green were considered unmarketable. Percentages by fresh weight of marketable, unmarketable, and rotten fruits were determined on fresh weight basis. In each plot, fresh weight of 100 marketable fruits was determined, and marketable fruit mean weight was calculated by dividing the fruit fresh weight by 100. The marketable fruit number was determined by dividing the marketable yield by the marketable fruit mean weight.

For the fruit quality assessment, 20 marketable fruits were used immediately after harvest for firmness determination on both sides of the equatorial zone using a digital penetrometer (T.R. Turoni s.r.l., Forl, Italy) with an 8 mm tip. The applied force for 4 mm penetration was expressed as kilogram-force (kgf). Twenty marketable fruits were used immediately after harvest for determining total soluble solids (TSS). Marketable fruits were homogenized in a blender (2 L; Waring HGB140—Warning Commercial, Stamford, CT, USA) for one minute at a low speed. The homogenate was filtered using a double cheesecloth and the TSS of the filtered juice were measured at 20 °C using a digital Atago N1 refractometer (Atago Co., Ltd., Tokyo, Japan). The TSS were expressed as °Brix.

*2.5. Economic Analysis*

Starting from the agronomic results of marketable yield and total soluble solids (TSS) of tomato fruits, the operating margin per hectare of processing tomato at the different levels of irrigation was calculated as indicator of profitability for the Italian market. Following the approach used by the Italian Farm Accountancy Data Network (FADN), the operating margin can be calculated by subtracting from gross sealable production (product sales, European Union (EU) direct production aids, possible changes in stocks value and product self-consumptions) the direct costs for cultivation inputs (commodities and services), as well as machine- and labour costs, assuming the latter two explicitly remunerated at the opportunity cost. The irrigation treatment that maximizes crop operating margin should be considered preferable.

In both years, marketable yield and TSS of tomatoes were estimated for the different irrigation treatments using the fit models developed in this study. Information on reference market prices of tomatoes (87 €/t for both years) were taken from price lists of the local Chamber of Commerce. According to the official differentiation criteria, price indices to be applied to reference market price, based on TSS (°Brix), were the following ones: 90% (4.00–4.49 °Brix), 95% (4.50–4.79 °Brix), 100% (4.80–5.29 °Brix), 105% (5.30–5.59 °Brix), 110% (5.60 °Brix and above), while fruit yield with TSS lower than 4.00 was considered unmarketable. On this basis, it was possible to determine the value of revenues from product sales under the different irrigation treatments in the two years, from which variable cultivation costs were detracted to obtain crop operating margins. It is worth to highlight that, with this method of calculation, the obtained operating margins do not account for the other components of gross sealable production (e.g., EU direct production aids, possible changes in stocks value and product self-consumptions), instead considered by the Italian FADN; however, these are not relevant for the purposes of this analysis.

Cultivation costs were estimated for the years 2017 and 2018, based on the records reported in the database of Italian FADN considering sample farms operating in the same

growing area where the trials were conducted. In particular, the processing tomato crop accounts of a constant sample of four farms in the two years were considered in a weighted average; variable cultivation costs, net of irrigation water costs, are reported in Table 1, properly modified, and integrated to consider the costs for purchasing irrigation driplines and for machine harvesting (the latter were issued by processing industry in the case of the sample farms).

**Table 1.** Variable production costs, net of the variable component of irrigation costs.

| Cultivation Inputs | Cost (€/ha) | |
|---|---|---|
| | 2017 | 2018 |
| Fertilizers | 641 | 593 |
| Pesticides | 818 | 553 |
| Seedlings | 756 | 795 |
| Irrigation dripline | 460 | 460 |
| Mechanization | 129 | 138 |
| Machine harvesting | 500 | 500 |
| Energy and other costs | 222 | 188 |
| Labour | 1274 | 1103 |
| Total | 4799 | 4329 |

Actual irrigation water volumes supplied under the different treatments were estimated based on the measurements of the local agrometeorological control unit of the Regional Agency for Agricultural Development and Innovation of Lazio (ARSIAL). Water volumes in 100% SMS-based treatment were considered in total 3605 m$^3$ in 2017 and 3040 m$^3$ in 2018, and thus, water volumes under the other irrigation treatments were proportionally calculated.

The unitary variable cost of irrigation water was determined based on FADN for processing tomato. Irrigation water cost under the different treatments was reported in Table 2.

**Table 2.** Variable costs of irrigation water in different irrigation treatments.

| Irrigation Level (% SMS Treatment) | Cost (€/ha) | |
|---|---|---|
| | 2017 | 2018 |
| 13.2 | 81 | 68 |
| 16.7 | 102 | 86 |
| 25.4 | 156 | 132 |
| 33.3 | 205 | 173 |
| 50.0 | 307 | 259 |
| 62.3 | 382 | 323 |
| 82.5 | 506 | 427 |
| 100.0 | 614 | 518 |
| 186.8 | 1147 | 968 |

For each year, variable cultivation costs under the different treatments can be derived by adding the cost component listed in the last row of Table 1 to the irrigation cost characterizing each treatment (Table 2).

### 2.6. Statistical Analysis

All data were subjected to two-way ANOVA using the SPSS22 software package (Chicago, IL, USA). Before analysis of variance, percentage data of marketable, immature, and rotten fruits were subjected to arcsine transformation to make the distribution normal. Percentage data were separated using Duncan multiple range test at a 5% level of significance. In each year, linear-plateau regression analysis was performed for investigating the relationship between crop traits (marketable and total yield, marketable fruit mean

weight, marketable fruit number) and irrigation levels while a linear regression analysis was performed for studying the relationship between marketable fruit quality traits, such as total soluble solids and firmness, and irrigation levels.

## 3. Results

### 3.1. Weather Data and Growing Degree Days

During 2017 growing period, the minimal temperature ranged between (10.1 and 23.4 °C), while the maximal temperature ranged between (22.1 and 36.5 °C). The rainfall period had a total amount of 16.7 mm. The highest rainfall amount (9.3 mm) was observed 45 DAT (Figure 2). In 2017, the accumulated value of growing degree days (GDD) from transplanting to harvest was 1208.9 °C. The total evapotranspiration ($ET_0$) value during this growing period was 486.2 mm.

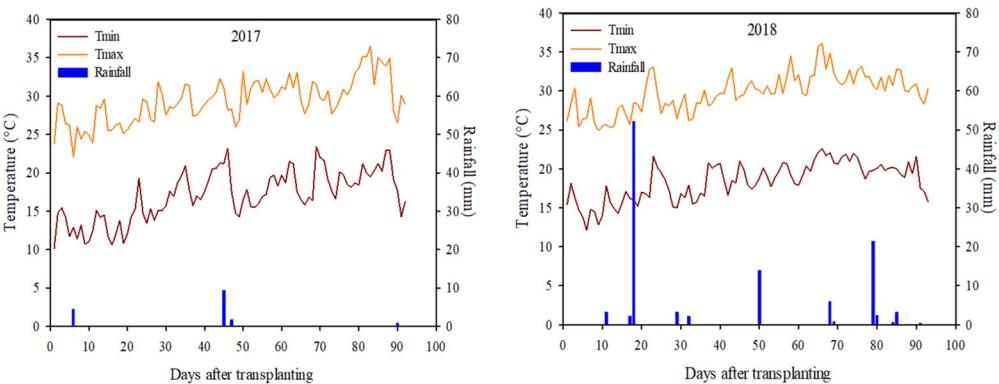

**Figure 2.** Daily maximum and minimum air temperature and rainfall during the 2017 and 2018 field trials of processing tomato.

During 2018 growing cycle, the minimal temperature ranged between (12.1 and 22.6 °C), and the maximal temperature ranged between (24.9 and 36.1 °C). The rainfall period had a total amount of 111.4 mm. The highest rainfall amount (52.1 mm) was observed at 18 DAT (Figure 2). The accumulated value of GDD from transplanting to harvest was 1307.9 °C while the total $ET_0$ value during this growing period accounted for 469.8 mm.

### 3.2. Yield and Fruit Quality

Total yield, percentage of marketable fruits, percentage of rotten fruits, marketable fruit number, marketable fruit mean weight, fruit firmness, and total soluble solids were significantly affected by year (Y). The irrigation level (I) influenced all tomato traits except the unmarketable yield. Except for marketable fruit mean weight and fruit firmness, all studied tomato traits were influenced by the Y × I interaction (Table 3).

**Table 3.** Analysis of variance for tomato traits recorded under nine irrigation levels during the two growing cycles.

| T | TY | MY | UM | PM | PI | PR | MF | FW | FF | TS |
|---|----|----|----|----|----|----|----|----|----|----|
| Y | * | ns | ns | *** | ns | *** | *** | *** | *** | *** |
| I | *** | *** | ns | *** | ** | *** | *** | *** | * | *** |
| Y × I | *** | *** | *** | *** | *** | *** | *** | ns | ns | * |

T = Treatment; Y = Year; I = Irrigation level; TY = Total yield; MY = Marketable yield; UM = Unmarketable yield; PM = Percentage of marketable fruits; PI = Percentage of immature fruits; PR = Percentage of rotten fruits; MF = Marketable fruit number; FW = Marketable fruit mean weight; FF = Fruit firmness; TS = Fruit total soluble solids. ns, *, **, *** Nonsignificant or significant at $p < 0.05$, 0.01, and 0.001, respectively.

According to the linear-plateau models (Figure 3), total and marketable yield increased linearly in both years with a more pronounced slope in 2017 (1.29 and 1.28 for total and marketable yield, respectively) in comparison to 2018 (0.60 for both total and marketable

yield). In 2017, total and marketable yields were maximized by supplying 92.8% and 96.2% of the irrigation level used in 100% SMS treatment, respectively. Moreover, 95.6% and 91.2% of the irrigation levels used in 100% SMS treatment were necessary in 2018 to maximize total and marketable yield, respectively (Figure 3). The estimated maximum total yield and marketable yield were higher in 2017 than 2018 (122.5 vs. 89.4 t/ha for total yield and 114.0 vs. 77.2 t/ha for marketable yield, respectively).

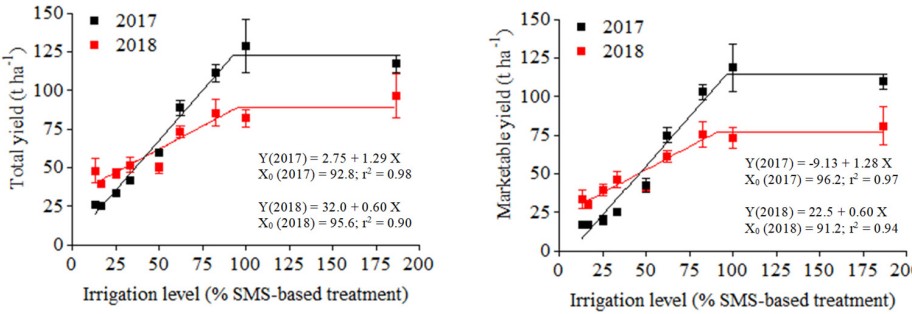

**Figure 3.** Effect of irrigation levels on total and marketable yield of processing tomato in 2017 and 2018. Data are means of three replicates ± standard errors. SMS = soil moisture sensor. Linear-plateau regression models for each year and crop traits are displayed in the figure.

The mean fruit weight increased linearly in both years up to 91.9% in 2017 and 118.9% in 2017 with a more pronounced slope in 2017 (0.45) compared to 2018 (0.25) (Figure 4). Similarly, fruit number increased linearly in both years up to 75.9% in 2017 and 82.4% in 2018 with a more pronounced slope in 2017 (2.19) compared to 2018 (0.6) (Figure 4).

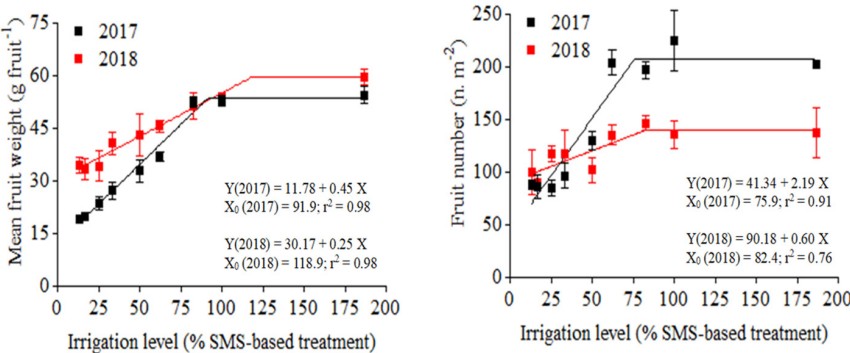

**Figure 4.** Effect of irrigation levels on mean weight and number of marketable tomato fruits in 2017 and 2018. Data are means of three replicates ± standard errors. SMS = soil moisture sensor. Linear-plateau regression models for each year and crop trait are displayed in the figure.

Unmarketable yield was significantly affected by the interaction Y × I (Table 3) with significant differences among irrigation levels only in 2018 where the highest unmarketable yield (15.4 t/ha) was recorded for the highest irrigation level (186.8% of SMS treatment) (Data not shown).

Figure 5 reports the percentage of marketable, immature, and rotten fruits for both growing cycles. In 2017, the percentage of marketable fruits was above 92% when the crop was irrigated with 82.5, 100 and 186.8% of the SMS based treatment while the lower irrigation volumes (13.2, 16.7, 25.4, 33.4% of SMS based treatment) resulted in the lowest percentage of marketable fruits (below 67%). Immature fruits and rotten fruits followed an opposite behavior with highest values under lower irrigation levels. In 2018, the differences among irrigation levels were less pronounced due to the higher rainfall amount (111.4 mm) in comparison to 2017 (16.7 mm). The percentage of marketable fruits was reduced only under severe irrigation deficit (13.2 and 16.7% of SMS-based treatment) and only the lowest irrigation level (13.2% of SMS-based treatment) caused the highest percentage of unmarketable (immature and rotten) fruits.

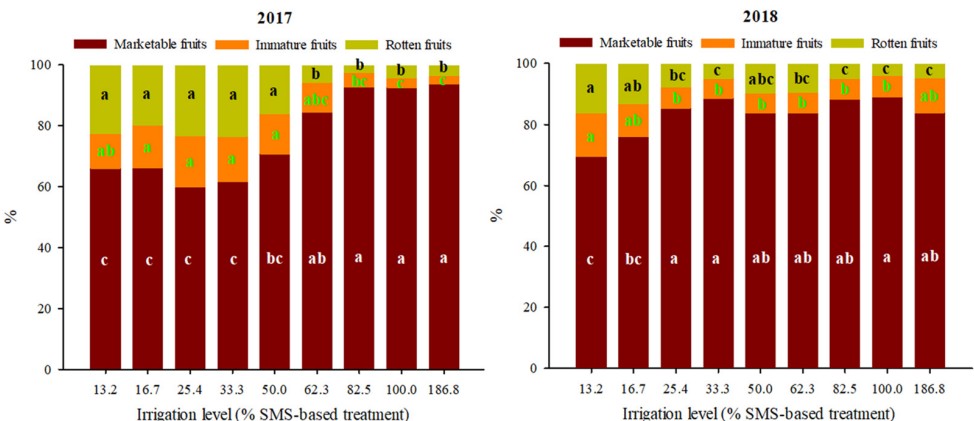

**Figure 5.** Effect of irrigation level on percentage by fresh weight of marketable, immature, and rotten fruits of processing tomato in 2017 and 2018. SMS = soil moisture sensor. Data are back transformed from arcsine transformation. In each year, different letters within each fruit group indicate significant differences according to Duncan's multiple range test ($p = 0.05$).

Fruit firmness outcomes was higher in 2018 compared to 2017 (3.6 vs. 1.05 kgf) while an increase of irrigation level caused a linear decrease of fruit firmness only in 2018 (Figure 6).

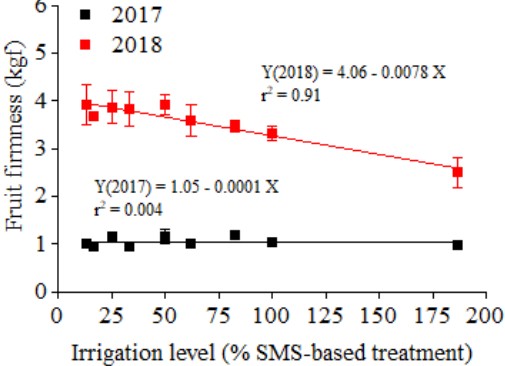

**Figure 6.** Effect of irrigation level on fruit firmness in 2017 and 2018. Data are means of three replicates ± standard errors. SMS = soil moisture sensor. Linear regression models for each year are displayed in the figure.

In both years, total soluble solids of fruit juice linearly decreased by increasing the irrigation level with a more pronounced effect in 2017 than in 2018 as demonstrated by the higher slope of 2017 than 2018 linear regression model (0.0179 vs. 0.0075) (Figure 7).

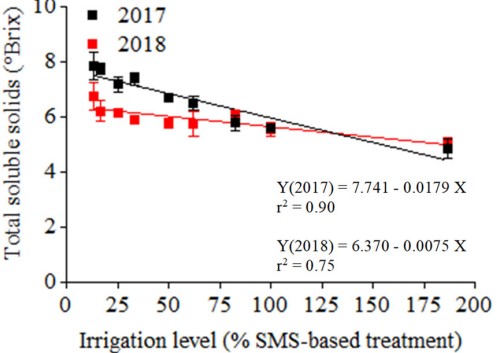

**Figure 7.** Effect of irrigation level on total soluble solids of tomato fruits in 2017 and 2018. Data are means of three replicates ± standard errors. SMS = soil moisture sensor. Linear regression models for each year are displayed in the figure.



### 3.3. Economic Outcomes

Table 4 summarizes the different components of the operating margin per hectare, quantifying their values in the two years and under the nine irrigation treatments applied in the field trials. Marked differences emerged between the two years, mainly due to seasonal weather conditions which were particularly drier in 2017 than 2018. This has extremized the differences among irrigation treatments, in terms of estimated marketable yields (ranging from 7.7 to 119.1 t/ha for 2017, and from 30.4 to 82.4 t/ha for 2018) and TSS of marketable fruits (between 4.40 and 7.51 °Brix in 2017, and from 4.97 to 6.27 °Brix in 2018). Market price had a very limited impact because, apart from the 186.8% SMS-based treatment, the values of TSS were always higher than 5.60 °Brix, the threshold above which a 10% increase in unitary price was applied (from 87 to 95.70 €/t). Based on the values of the operating margins at different irrigation levels (Table 4), 100% SMS-based treatment should be preferred since it led to the best results in both years.

**Table 4.** Revenues, variable costs, and operating margins at the different levels of irrigation in 2017 and 2018 tomato growing cycle.

| Irrigation Level (% of SMS Treatment) | Revenues from Product Sales (€/ha) | | Variable Costs (€/ha) | | Operating Margins (€/ha) | |
|---|---|---|---|---|---|---|
| | 2017 | 2018 | 2017 | 2018 | 2017 | 2018 |
| 13.2 | 740 | 2909 | 4880 | 4397 | −4140 | −1488 |
| 16.7 | 1171 | 3110 | 4902 | 4415 | −3731 | −1305 |
| 25.4 | 2247 | 3613 | 4956 | 4461 | −2709 | −848 |
| 33.3 | 3215 | 4065 | 5004 | 4502 | −1789 | −437 |
| 50.0 | 5260 | 5019 | 5106 | 4588 | 154 | 431 |
| 62.3 | 6767 | 5723 | 5182 | 4652 | 1585 | 1071 |
| 82.5 | 9243 | 6879 | 5306 | 4756 | 3937 | 2123 |
| 100.0 | 11,395 | 7884 | 5413 | 4847 | 5981 | 3036 |
| 186.8 | 9323 | 7167 | 5947 | 5297 | 3376 | 1870 |

Linear regression analysis between operating margin and irrigation level (from 100% to 13.2% of SMS treatment) yielded negative operating margins when the irrigation level was below 48.7% in 2017 and 41.7% in 2018 (data not shown).

## 4. Discussion

Among the two growing cycles, the 2017 was drier than 2018 because there were 6 days of rainfall with the highest rainfall amounts recorded on 6 and 45 days after transplanting (DAT) while in 2018, there were 16 days of rainfall. The highest rainfall amount in 2018 was recorded on 18 DAT, followed by 79 DAT and 50 DAT. Moreover, total $ET_0$ value was highest in 2017 growing cycle.

In 2018 growing season, the heavy rainfalls caused an increase in the duration of crop cycle as demonstrated by the higher growing degree days required from transplanting to harvest in comparison with 2017 growing season (1307.9 vs. 1208.9 °C).

Total and marketable yields were higher in 2017 than in 2018 (Figure 3) due to the physical damage of leaf surface and the increase of flowers drop-off occurring in 2018 growing season as a result of the heavy precipitations at 18 (52.1 mm in few hours) and at 50 days after transplanting (13.9 mm in few hours). This outcome is in line with findings of some authors who observed that heavy rainfalls during flowering and fruit growth lowered tomato yield due to the drop-off of flowers and fruit damage [31,32]. In the current trial, the negative effects of the two intensive rainfalls occurring in 2018 growing season on flowers and fruit setting was confirmed by the lower fruit number recorded in 2018 in comparison with 2017 (Figure 4).

In both growing cycles, the effect of irrigation level on marketable yield was related to the change of both fruit number and fruit mean weight (Figure 4). Moreover, the 2017–2018 research findings indicated that a lower amount of irrigation water was required

for maximizing the fruit number (75.9% of SMS treatment in 2017; 82.4% of SMS treatment in 2018—Figure 4) in comparison with fruit mean weight (91.9% of SMS treatment in 2017; 118.9% of SMS treatment in 2018—Figure 4). The driest growing cycle (2017) led to a more pronounced increase in mean fruit weight by raising the irrigation level as demonstrated by the slope of linear regression in comparison to 2018 growing cycle (0.45 vs. 0.25—Figure 4).

The higher percentage of unmarketable fruits (immature and rotten fruits) recorded with the lower irrigation regimes can be explained by the susceptibility of less irrigated plants to several physiological disorders in their fruits such as the blossom-end rot and yellow shoulder tomato phenomena in their fruits. The blossom-end rot can occur when tomato plants have a decreased ability to internally translocate calcium toward fruits during drought circumstances [33]. The yellow shoulder tomato disorder is due to several factors, one of them is the frequent exposure to the sun's direct rays, where the shoulders of tomatoes frequently acquire yellowing due to high temperatures and direct sunlight, which also reduces the amount of the tomato fruit's red pigment lycopene [34]. In this study, both phenomena may have occurred and especially in 2017 due to more water shortage caused by less rainfall amount leading to a lack of fruit coverage by leaves. Some authors also indicated that a water deficit affects the plant photosynthesis resulting in a reduction of leaf area [35].

The capacity of fruits to be mechanically harvested, transported, and preserved for a long time depends heavily on their firmness since soft fruits are more vulnerable to mechanical damage and bacterial or fungal infections that cause fruit loss [36]. In this study, the decrease in fruit firmness with the increase of water levels in 2018 was consistent with many other studies [37,38]. Several authors have suggested that variations in cell turgor and epidermal wall flexibility are responsible for the differences in fruit firmness caused by plant water stress [39–41]. The lower firmness value obtained in 2017 in comparison to 2018 can be explained by the increase of the activity of degrading enzymes during tomato ripening [42]. In this study, in the last 10 days before harvest the maximum temperature in 2018 was significantly lower than 2017, where temperature peaks were observed above 35 °C (Figure 2). This may have affected the softening of the fruits by influencing the activity of cell wall degrading enzymes (pectinestrase, polygalacuronase, and cellulase). As reported by [43], the activities of partially purified degrading enzymes polygalacturonase and cellulase from tomato fruits reached maximum values under high temperatures (about 30 °C for the activity of polygalacturonase and about 35 °C for the activity of cellulase). The above findings can explain the lower fruit firmness recorded in 2018 in comparison to 2017.

The TSS represent the percentage of dissolved solids in a solution, and it is the most significant fruit quality criterion for both fresh market and processed tomatoes [44]. Soluble solids are mainly represented by soluble carbohydrates and organic acids, which are the basic components of fruit flavor and taste [45]. The TSS diminution in tomato fruits in parallel with augmentation of irrigation levels during both years was probably due to a lower water fruit content [46] and to a higher accumulation of sugars in fruits [47].

Results of the economic analysis showed that the currently adopted Italian criteria of price differentiation based on TSS are not adequate for promoting further improvement of fruit quality at the expense of yield, if the TSS level is already high (as in the case of the production obtained from the field trials conducted in this study). From the economic point of view, these research findings demonstrated that in absence of limitations on water availability and given the relatively limited incidence of the costs for irrigation water it is not convenient to adopt water management strategies based on deficit irrigation, that reduce yield and operating margin at irrigation levels below 100% SMS-based treatment. This is in line with the findings of [22] considering the impact of deficit irrigation on net income and explains why this approach to water management finds its main application to face conditions of water scarcity [48]. Instead, overirrigation resulted in a decrease of operating margin in both growing cycles due to the deterioration of fruit quality in absence of any yield gain, confirming the findings of [49], and to the increase of irrigation cost.

## 5. Conclusions

Monitoring soil moisture content is essential to manage the irrigation, maximize marketable yield, fruit quality and farmer income. The outcomes of this study showed that irrigation level close to 100% of water restitution based on the low-cost wireless soil moisture capacitance sensor (Precision™ Soil Sensor—Toro, Bloomington, MN, USA) placed halfway between plants along the rows resulted in the optimization of operating margins for the Italian market due to the maximization of marketable fruit yield without detrimental effect on fruit quality (TSS) in both growing years. On the other hand, irrigation levels below 100% of water restitution led to a decrease of operating margins due to a reduction of marketable yield with a slight increase of TSS under reduced irrigation regime. Similarly, irrigation levels above 100% of water restitution resulted in a reduction of operating margins due to a decrease of TSS and an elevation of irrigation costs without any benefit in terms of marketable yield increase.

**Author Contributions:** Conceptualization, M.C., P.S., R.M. and G.C.; methodology, M.C., P.S., R.M., R.C. and G.C.; validation, M.C., A.E.C., R.M., R.C., D.D. and G.C.; formal analysis, M.C., R.M., R.R., R.C. and G.C.; investigation, M.C., A.E.C., R.M., R.R., R.C., D.D. and G.C.; resources, R.M., R.R. and G.C.; data curation, M.C., A.E.C., R.M., R.R., R.C., D.D. and G.C.; writing—original draft preparation, M.C., A.E.C., R.C., D.D. and G.C.; writing—review and editing, M.C., A.E.C., R.C., D.D. and G.C.; visualization, M.C., A.E.C. and G.C.; supervision, M.C., R.M. and R.R.; project administration, R.M. and R.R.; funding acquisition, R.M. and R.R. All authors have read and agreed to the published version of the manuscript.

**Funding:** This work was partially funded by MIUR in the frame of the initiative "Departments of excellence", Law 232/2016. This work was also partially funded by ARSIAL in the frame of ARSIAL-DAFNE Agreement.

**Data Availability Statement:** Data available on request from the corresponding author.

**Acknowledgments:** We thank Silvano Di Giacinti for technical support in conducting the field trials at the Experimental Farm of ARSIAL. We also thank the TORO System Europe for providing the irrigation lines, soil moisture sensors and controllers.

**Conflicts of Interest:** Piero Santelli from Toro System Europe was involved in the conceptualization and methodology set up of the trials without any role in the collection, analyses, interpretation of data, in the writing of the manuscript, and in the decision to publish the results.

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
