# Peer review of "Optimization of Applied Irrigation Water for High Marketable Yield, Fruit Quality and Economic Benefits of Processing Tomato Using a Low-Cost Wireless Sensor"

_horticulturae, doi:10.3390/horticulturae9030390_

Round 1
Reviewer 1 Report
1.The irrigation treatments were obtained using 9 drip lines having 9 different water flow rates from 1.5 to 21.3L/m, There is no specific explanation in the text whether the spacing between the nine drip lines droppers is consistent. Due to the large difference in flow rates, there will be a large difference in the wetted area in the field, resulting in differences in evaporation and affecting the experimental results.
2.What is the ET calculation formula in the text? Different calculation formulas can have significant differences.
3.There is no specific explanation in the text on how to obtain the field capacity, which may affect the results in the text.
4.Why is the sensor buried at a depth of 10cm? the effective water absorption depth of tomato roots is 10 cm? Would different burial depths lead to different results?
5.The article does not explain how fertilizers are applied. If fertilization is done through drip irrigation, the amount of irrigation water will also affect the amount of fertilizer applied and its effectiveness, which may have an impact on the results of this study.
6.The term 'optimization' used in the title of this article is based on data from only two years, which is not sufficient to support the purpose of optimization. It is recommended to change it to 'evaluation'.
Author Response
1.The irrigation treatments were obtained using 9 drip lines having 9 different water flow rates from 1.5 to 21.3L/m, There is no specific explanation in the text whether the spacing between the nine drip lines droppers is consistent. Due to the large difference in flow rates, there will be a large difference in the wetted area in the field, resulting in differences in evaporation and affecting the experimental results.
Dear Reviewer, let me draw you attention that the soil contained a large proportion of sand (48%) providing a wetting pattern which was much more vertical than horizontal due to the pull of the gravity. This should have limited differences in wetting area of soil surface among irrigation levels. Moreover, because processing tomato is a fast-growing crop, evaporation decreases quickly over the growing period as the crop develops and the crop canopy shades more and more of the ground area. For the above reasons we can expect that differences in evaporation among treatments were negligible.
2.What is the ET calculation formula in the text? Different calculation formulas can have significant differences.
The estimation of ET was obtained using the Pan evaporation method as follow.
ETo = Kp Epan
where
ETo reference evapotranspiration [mm/day],
Kp pan coefficient,
Epan pan evaporation [mm/day] from a Stainless Steel Class A Evaporation Pan sited on a green grass cover.
The above findings have been inserted in the revised version of manuscript (L109-113).
3.There is no specific explanation in the text on how to obtain the field capacity, which may affect the results in the text.
Dear Reviewer, let me draw you attention that the field capacity was determined using a built-in calibration process as proposed by Toro Company and used in other research studies (Serena et al., 2020). Following the manufacturer’s instructions, near-saturation soil conditions were created by running two cycles of irrigation and wetting the soil around the sensors and, then, the calibration process was initiated. The soil moisture sensor system defined field capacity based on the soil moisture values recorded after 4 h of drainage following the saturation event (Serena et al., 2020*).
The above findings have been inserted in the revised version of manuscript (L141-147).
*Serena, M.; Velasco‐Cruz, C.; Friell, J.; Schiavon, M.; Sevostianova, E.; Beck, L.; Sallenave, R.; Leinauer, B. Irrigation Scheduling Technologies Reduce Water Use and Maintain Turfgrass Quality. Agron. J. 2020, 112, 3456–3469, doi:10.1002/agj2.20246.
4.Why is the sensor buried at a depth of 10cm? the effective water absorption depth of tomato roots is 10 cm? Would different burial depths lead to different results?
Dear Reviewer, the goal of the study was to test a simple and economic moisture sensor having probes 10 cm long. We followed the manufacturer’s instructions for installation. We decided to use this type of sensor because as reported by Oliveira et al. (1996*), root length intensity of drip-irrigated tomato plants showed an exponential decrease with increasing depth providing similar root intensity at 10 cm intervals from soil surface up to 40 cm depth.
*Oliveira et al. (1996). Tomato Root Distribution under Drip Irrigation. J. AMER. SOC. HORT. SCI. 121(4):644–648.
5.The article does not explain how fertilizers are applied. If fertilization is done through drip irrigation, the amount of irrigation water will also affect the amount of fertilizer applied and its effectiveness, which may have an impact on the results of this study.
Dear Reviewer, we apologize for missing the information on fertilization program. A slow-release mineral fertilizer containg 16N-9.6P-13.2K was pre-plant applied at a rate of 1 t/ha followed by two side dress applications during flowering of a granular ammonium nitrate fertilizer (26%N) at a rate of 76.9 kg/ha for each application. Granular fertilizer was mixed into the upper portion of the soil just after side dress application. The above information has been inserted in the revised version of manuscript (L131-135).
6.The term 'optimization' used in the title of this article is based on data from only two years, which is not sufficient to support the purpose of optimization. It is recommended to change it to 'evaluation'.
The title has been changed according to your comment.
Reviewer 2 Report
The manuscript titled "Optimization of applied irrigation water for high marketable yield, fruit quality and economic benefits of processing tomato using a low-cost wireless sensor" submitted by Chami et al. the manuscript over all very interesting and writen very well. but before acceptance, I have some minor corrections.
Section introduction line 72: (Solanum lycopersicum) should be italic
Section Introduction line reomove typo,, m−1, power should be super scipt
Section material, heading "Experimental design and treatments" line136: SMS? define full form at first use
Section material, heading "Experimental design and treatments" line209: EU define full form at first use
section disussion, line 41 "Total soluble solids (TSS)" only define one time,,
remove all typo from manuscript,
revise the manuscript eglish,, try to avoid from t=repitation..
Author Response
The manuscript titled "Optimization of applied irrigation water for high marketable yield, fruit quality and economic benefits of processing tomato using a low-cost wireless sensor" submitted by Chami et al. the manuscript over all very interesting and written very well but before acceptance, I have some minor corrections.
Dear Reviewer, thank for considering the manuscript very interesting and written very well and for providing useful corrections.
Section introduction line 72: (Solanum lycopersicum) should be italic
Done
Section Introduction line reomove typo,, m−1, power should be super scipt
Done
Section material, heading "Experimental design and treatments" line136: SMS? define full form at first use
Done
Section material, heading "Experimental design and treatments" line209: EU define full form at first use
Done
section discussion, line 41 "Total soluble solids (TSS)" only define one time, remove all typo from manuscript
Done
revise the manuscript English,, try to avoid from repetition
Done
Reviewer 3 Report
The manuscript is written with clear understanding of the project addressed. However, there are major concerns that need to be addressed to enhance the quality of the manuscript. My specific comments are as follows:
Abstract: Add concluding remark.
Introduction:
Based on your objectives, please compare how your study is different from those that have already been published.
Materials and Methods:
There is no information regarding the sample preparation eg. Storage/fruit samples/variety
Results and Discussion:
Section 3.1: Is there significant difference between dataset for both 2017 and 2018? Discuss
I suggest add the mean and standard deviation between the parameters to show the comparison. Table 3 only shows the significant difference at p<0.05 without discussing further about the mean values in details
Is there any difference for total yield and marketable yield in Figure 3? Clearly, we can see that the yields for 2017 are higher than 2018. The authors need to discuss on this matter
L342: “Fruit firmness outcomes was higher in 2018 compared to 2017 (3.6 vs 1.05 kgf) while an increase of irrigation level caused a linear decrease of fruit firmness (y=2.557-0.004x; r2=0.81) (Data not shown).” If data was not shown, better omit this sentence. If you mention in the text, make sure to include the respective figure/table
Same comment with the data not shown for the next sentence.
L355-359: “Fruit pH was significantly affected by Y x I interaction (Table 3). No significant difference…” should be placed under section 3.2
Relate your results with existing literatures to support your findings. Instead of mentioning the results, the authors should justify/explain the findings
Conclusion:
Too short. Elaborate more as suggested below
Add on main finding/results of the study. What are the main outcome based on the results. The authors should highlighted this matter.
General comments:
Please check the reference styles and grammar of the manuscript.
Author Response
The manuscript is written with clear understanding of the project addressed. However, there are major concerns that need to be addressed to enhance the quality of the manuscript.
Dear Reviewer, thank for considering the manuscript with clear understanding of the project addressed and for providing useful corrections.
Abstract: Add concluding remark.
Dear Reviewer, we have already indicated the following concluding remarks in the abstract (L28-31).
Economic analysis demonstrated that 100% of irrigation level should be preferred by the Italian farmers since it maximized the operating margins of processing tomatoes in both years. To con-clude, the use of the low-cost wireless soil moisture sensor is an effective tool to manage the level of irrigation and optimize the processing tomato yield and economic benefits for farmers.
Introduction:
Based on your objectives, please compare how your study is different from those that have already been published.
Dear Reviewer, let me draw your attention that in the introduction we cited the works conducted using soil moisture sensor for irrigation scheduling of processing tomato such as:
[26] Madramootoo, C.A.; Jaria, F.; Arumugagounder Thangaraju, N.K. Irrigation Scheduling and Requirements of Processing To-mato (Lycopersicon Esculentum L.) in Eastern Canada. Irrig. Sci. 2021, 39, 483–491, doi:10.1007/s00271-021-00731-5.
[27] Vázquez, N.; Huete, J.; Pardo, A.; Suso, M.L.; Tobar, V. Use of Soil Moisture Sensors for Automatic High Frequency Drip Irrigation in Processing Tomato. Acta Hortic. 2011, 229–235, doi:10.17660/ActaHortic.2011.922.30.
We also highlighted that the previous studied were limited to one irrigation level and one-growing cycle without deep investigation of the effects of different irrigation levels and growing seasons on agronomic and economic aspects of processing tomato. As stated in the introduction the goal of the study was also to test a simple and economic moisture sensor for irrigation scheduling of processing tomato.
Materials and Methods:
There is no information regarding the sample preparation eg. Storage/fruit samples/variety
Dear Reviewer, we used the tomato variety ‘Perfectpeel’ (ex PS 1296) from Bayer Seminis Milano, Italy (L127-128). Concerning the yield measurements and fruit quality assessments, we did not store any fruits because the measurements/analysis were made immediately on fresh fruits after harvest in a laboratory located in the same Research Station where the experiment was conducted (L207; L211).
The sample sizes were:
- central double row of each plot for determining yield (L199);
- 100 marketable fruits per plot for determining marketable fruit mean weight (L204-205);
- 20 marketable fruits per plot to measure fruit firmness (L207-208);
- 20 marketable fruits per plot to determine soluble solids and pH (L210-211).
Results and Discussion:
Section 3.1: Is there significant difference between dataset for both 2017 and 2018? Discuss
Dear Reviewer, as you can see from the Table 3, the interactions between year and irrigation level were significants for all crop traits except for fruit firmness and marketable fruit mean weight. For the above reasons, we discussed the interaction effects of the two factors using the plateau-linear or linear models.
I suggest add the mean and standard deviation between the parameters to show the comparison. Table 3 only shows the significant difference at p<0.05 without discussing further about the mean values in details
Dear Reviewer, let me draw you attention that we already inserted the standard error bars in the figures as indication of variability of sample means. Because there were significant interactions between factors, we fitted regression models separately for each year to study the changes of crop traits across the irrigation levels; we believe that this approach provides more information about the relationship between the two variables in comparison with mean separation tests. Moreover, the results were displayed in figures and they cannot be duplicated in the table 3.
Is there any difference for total yield and marketable yield in Figure 3? Clearly, we can see that the yields for 2017 are higher than 2018. The authors need to discuss on this matter.
Dear Reviewer, we indicated the maximum values of total and marketable yield for each year (L315-317); these results were discussed in L410-418.
L342: “Fruit firmness outcomes was higher in 2018 compared to 2017 (3.6 vs 1.05 kgf) while an increase of irrigation level caused a linear decrease of fruit firmness (y=2.557-0.004x; r2=0.81) (Data not shown).” If data was not shown, better omit this sentence. If you mention in the text, make sure to include the respective figure/table
Dear Reviewer, we added a new figure (n. 5) to show the fruit firmness results.
Same comment with the data not shown for the next sentence.
The sentence has been removed and we also removed in the materials and methods the determination of juice pH.
L355-359: “Fruit pH was significantly affected by Y x I interaction (Table 3). No significant difference…” should be placed under section 3.2
The sentence has been removed and we also removed in the materials and methods the determination of juice pH.
Relate your results with existing literatures to support your findings. Instead of mentioning the results, the authors should justify/explain the findings
Dear Reviewer, we discussed the results considering the existing literature. We explain the reasons of yield variation across the year considering the weather conditions (L416-424). Moreover, we related the yield variations to yield components and we linked the yield components to weather conditions (L425-432). Differences on fruit partitioning among categories (marketable, rotten , etc.) were discussed considering the weather data and the irrigation level (L434-446). Fruit firmness and TSS were linked to weather condition and irrigation level (L446-470). Finally the economic results were explained considering the operational margin resulting from the impact of irrigation level on marketable yield and fruit quality (TSS).
Conclusion:
Too short. Elaborate more as suggested below
Add on main finding/results of the study. What are the main outcome based on the results. The authors should highlighted this matter.
Dear Reviewer, we elaborated the conclusions according to your suggenstions (L492-500).
General comments:
Please check the reference styles and grammar of the manuscript.
Dear Reviewer we checked the styles and grammar of the manuscript.
Round 2
Reviewer 3 Report
The authors have addressed all the comments. Hence, the paper can be accepted.